# Low-Phase-Noise Oscillator Using a High-$Q_L$ Resonator with Split-Ring Structure and Open-Loaded T-Type Stub for a Tumor-Location-Tracking Sensor

**Ki-Cheol Yoon** [1,2,†], **Kwang-Gi Kim** [1,2,3,4,*], **Jun-Won Chung** [5,6,*] **and Byeong-Soo Kim** [1,7,†]

1   Medical Devices R&D Center, Gachon University Gil Medical Center, Incheon 21565, Korea; kcyoon98@gachon.ac.kr (K.-C.Y.); qudtngh@snu.ac.kr (B.-S.K.)
2   Department of Biomedical Engineering, College of Medicine, Gachon University, Incheon 21565, Korea
3   Department of Biomedical Engineering, College of Health Science, Gachon University, Incheon 21936, Korea
4   Department of Health Sciences and Technology, Gachon Advanced Institute for Health Sciences and Technology (GAIHST), Gachon University, Incheon 21565, Korea
5   Department of Gastroenterology, College of Medicine, Gachon University, Incheon 21565, Korea
6   Department of Gastroenterology, Gachon University Gil Hospital, Incheon 21565, Korea
7   Interdisciplinary Program in Bioengineering, Graduate School, Seoul National University, Seoul 08826, Korea
*   Correspondence: kimkg@gachon.ac.kr (K.-G.K.); drgree@gilhospital.com (J.-W.C.); Tel.: +82-10-3393-4544 (K.-G.K.); +82-10-9004-1604 (J.-W.C.)
†   Ki-Cheol Yoon and Byeong-Soo Kim contributed equally to this work and are co-first (lead) authors.

**Abstract:** Sensors in the medical field to detect specific tissues, such as radars, must provide accurate signals from frequency generators. In order to supply an accurate frequency signal, the oscillator must have a low phase noise. Therefore, the resonator used in the oscillator must provide a high $Q_L$. Therefore, in this paper, we have proposed a low-phase-noise X-band oscillator that used a resonator with a high value of $Q_L$ as a sensor for tissue-locating applications. The resonator had a split-ring structure and consisted of an open-loaded, T-type stub with a high-$Q_L$; such high-$Q_L$ levels were enabled by controlling the length of the open-circuit in the T-type stub. This led to the generation of only low-phase noise in the proposed oscillator. Experimental results showed that, at an operating frequency of 10.08 GHz, the output power was 18.66 dBm, the second harmonic suppression was −34.40 dBc, and the phase noise was −138.13 dBc/Hz at an offset of 100 kHz. This proposed oscillator can be used as a sensor to detect the location of tissues during laparoscopic surgery.

**Keywords:** high loaded quality factor; low phase noise; split-ring structure; open-loaded T-type stub; tissue location sensor

## 1. Introduction

The development of medical technology has seen rapid growth in the use of radio frequency (RF) applications in medical services [1]. Medical device systems that use microwaves are extensively employed in treatment and diagnostics [2–6]. In the former, they are used to treat inflammation and tumors, while in the latter, they aid in the detection of tissue-based substances, such as tumor cells, DNA, cortisol, and glucose [2–6]. Markers are substances that help to track the location of a tissue (such as a tumor) [1,7,8]. Since sensors work by generating electrical signals of the required frequency, oscillators are employed in marker sensors. The oscillator used by the sensor to supply the correct frequency signal must have low phase noise. A method for designing a low-phase-noise oscillator can be a resonator having a high $Q_L$ for the oscillator. Therefore, the design of a resonator with a high $Q_L$ is very important.

Dielectric resonators (DRs) are promising elements because of their high-$Q_L$ characteristics. However, their three-dimensional (3-D) structures not only limit their implementation to system-on-chip (SoC) and integrated circuit (IC) systems, but also render them unsuitable for mass production [9,10].

Resonators with planar structures (microstrip lines) are being extensively studied, with a specific focus on increasing their $Q_L$. The representative microstrip-line resonator is a hairpin structure, which has a physical length of $\lambda_g/2$. While hairpin structures are easy to fabricate [11], their $Q_L$ is too low in performance to apply them to an oscillator. To overcome this disadvantage, spiral resonators have been considered as an alternative. However, although they are small and have higher $Q_L$ values, they are still insufficient for application in oscillators [11].

To further increase the $Q_L$, a substrate-integrated waveguide (SIW) structure and an open-loop resonator with a T-stub inserted into it were studied in 2014. An analysis of their characteristics showed significantly improved values of $Q_L$. The SIW structure is patch-shaped and the structural characteristics of its feeding-line part make the $Q_L$ adjustable [12–15]. The $Q_L$ is increased by virtue of the dual structure used in the state where the T-stub is applied [16]; however, it also has the undesired effect of enlarging the resonator. This increase in area means that applying it to the oscillator would hinder system integration.

A triangle-folded resonator with a coupled line has a smaller size, with a physical length of $\lambda_g/4$ [17]. Since resonators do not use vias, this one has excellent performance because there is no energy loss to the via. However, although this device enabled the oscillator to achieve a low phase noise, its $Q_L$ was relatively low.

In this paper, a low-phase-noise oscillator with a high-$Q_L$ resonator, using a combination of a split-ring structure and an open-loaded T-type stub, is presented. The resonator is a combination of a split-ring structure and an open-loaded T-type stub, where a high-$Q_L$ characteristic can be obtained by adjusting the length of the open-loaded T-type stub. The characteristic of the resonator is the band-stop type of resonance ($S_{21}$). Owing to the sharp band-rejection characteristic of the new resonator, very low phase noise can be obtained for the oscillator. This design strives to obtain excellent performance so that it can be applied to tissue- and tumor-tracking sensors.

## 2. Tumor-Tracking Oscillator and Sensor

During laparoscopic surgery, tumors located inside the colon and stomach are difficult to locate in the cavity due to the peritoneum. This brings with it the risk of making incisions in the wrong locations during tumor extraction, which could lead to medical accidents. The X-band tissue search radar is also used to determine the location of the tumor, as shown in Figure 1 [18,19].

Radars utilize oscillators to generate detection frequencies; such oscillators require high-$Q_L$ resonators and low phase noise. Existing resonator-based sensors used in tissue detection have $Q_L$ values of at least 50, as shown in Table 1. However, when these resonators are applied to oscillators, the phase noise of the signal generated during abnormality detection does not easily drop below even $-100$ dBc/Hz at a 100 kHz offset [20,21]. Therefore, resonators with higher $Q_L$ values are needed to further reduce the phase noise of oscillators; this need is amplified when considering the possibility of the generation of severe jitter noises on the radar, which would hinder the detection of the marker.

**Table 1.** Resonator $Q_L$ for tissue marker applications.

| Reference | Resonant Frequency (GHz) | $Q_L$ | Detection Tissue |
|:---:|:---:|:---:|:---:|
| [4] | 11.2 | 100 | cortisol |
| [5] | 12.3 | 50 | DNA |
| [6] | 7.54 | 130 | glucose |

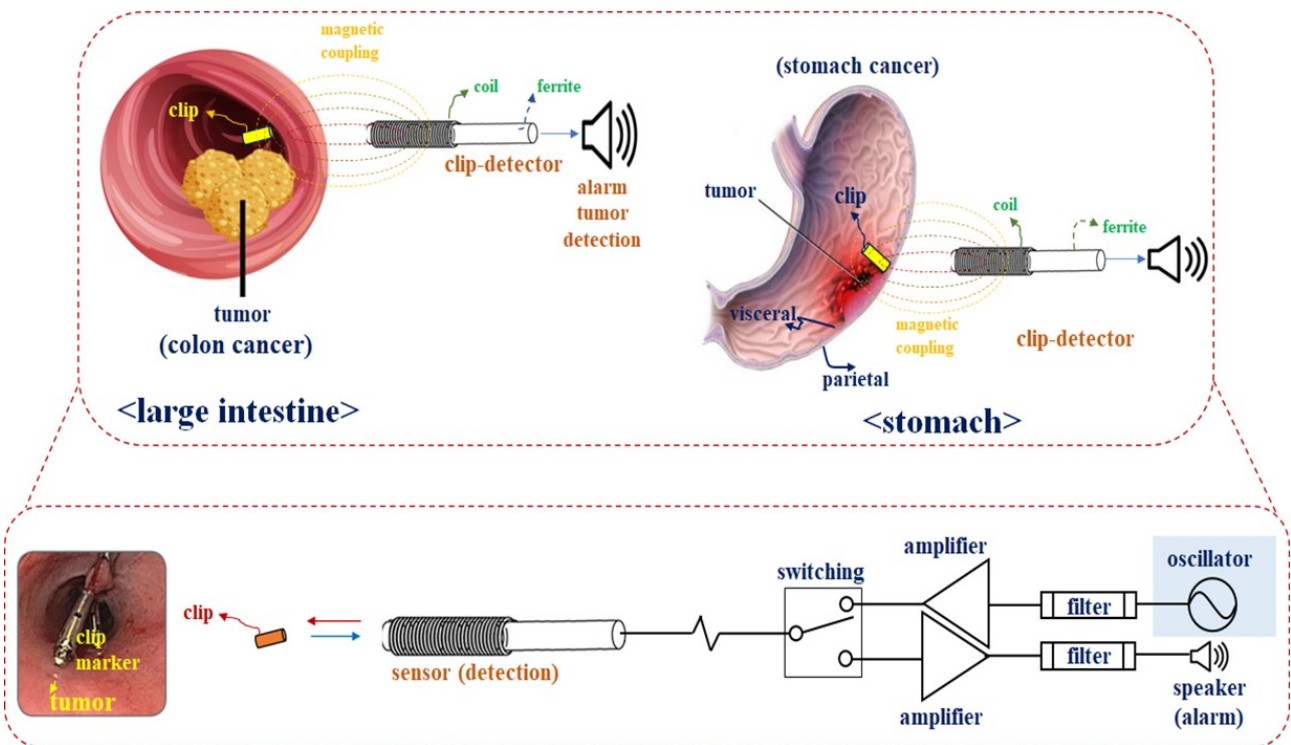

**Figure 1.** Oscillator and sensor for tumor tracking.

## 3. Design of a Resonator and Experimental Results

The proposed resonator is composed of a dual spilt-ring structure and an open-loaded T-type stub, as shown in Figure 2. From the figure, $Z_{ex}$ and $Z_{in}$ are the characteristic impedances of the external and internal split-ring structures, while $Z_T$ and $Z_s$ are those of the series transmission line and open-loaded stub in the T-type structure.

$Z_{p1}$ and $Z_{p2}$ are the characteristic impedances of the gaps represented by gap sizes $g_{ex}$ and $g_{in}$, while $Z_{op}$ is that of an open-loaded T-type stub. In addition, $l_{TH}$, $I_{TV}$, $l_{ex1}$, and $l_{ex2}$ are the physical lengths of the external split-ring structure, and $l_{in,m,n}$ ($m = 1$ to 3, $n = 4$ to 5) refers to the set of physical lengths of the internal split-ring structure. $l_T$ and $l_s$ are the physical lengths of the series transmission line and open-loaded stub in the T-type structure, respectively. The symbols $w_{ex}$, $w_{in}$, $w_T$, and $w_s$ denote the widths of the external and internal split-ring structures, the series transmission line, and the open-loaded stub in the T-type structure, respectively, while $g_{ex}$ and $g_{in}$ indicate the coupling gaps between the split-ring structures and the open-loaded T-type stub, respectively. $\theta_{TH}$, $\theta_{TV}$, $\theta_{ex1}$, and $\theta_{ex2}$ are the electrical lengths of the external split-ring resonator and $\theta_{in,m,n}$ ($m = 1–3$, $n = 4–5$) is that of the internal split-ring structure, while $\theta_T$ and $\theta_s$ are those of the series transmission line and open-loaded stub in the T-type structure. In the equivalent circuit depicted in Figure 2, $L_{ex}$ and $L_{in,m,n}$ ($m$: 1–3, $n$: 4–5) are the inductances of the external and internal split-ring structures. Then, $1/L_{in}$ can be expressed as $1/L_{in} + 1/L_{in} = L_{in}$ (for in for in, $m$: 1 to 3, $n$: 4 + 5). $L_T$ and $L_s$ are the inductances of the series transmission line and open-loaded stub, respectively, while $C_{ex}$, $C_{in}$, $C_T$, and $C_S$ are the capacitances of the external and internal split-ring structures, the series transmission line, and the open-loaded stub, respectively. $C_{op}$ is the mutual capacitance between the transmission line and the ground plane, with $C_{p1}$ and $C_{p2}$ being those of the coupling gap between the split-ring structure and the open-loaded T-type stub, respectively. Lastly, port 1 and port 2 are the input and output ports, and the characteristic impedance of the ports is 50 Ω, which is called the feeding-line in the resonator structure of Figure 2. Then, the gm is the gap size between the resonator and feeding line. The physical structure of $C_{p2}$ (see equivalent circuit) is carried out the same as with $g_m$.

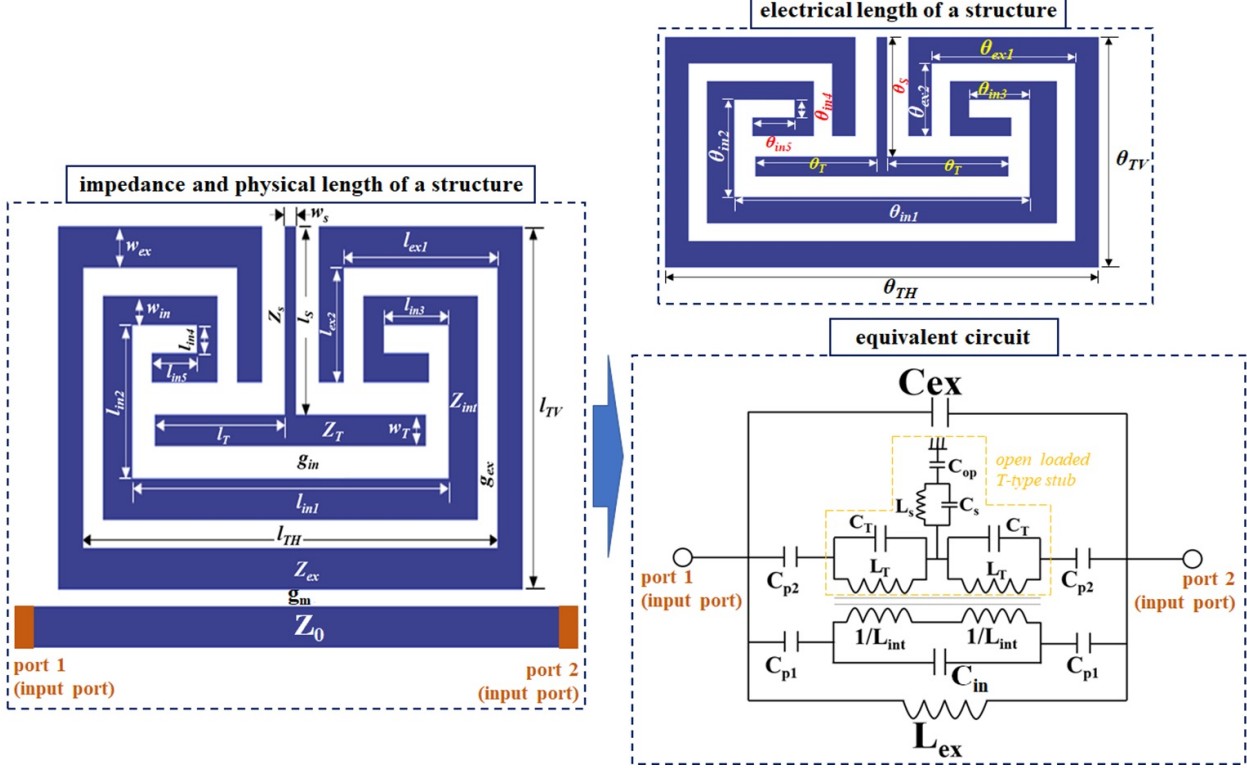

**Figure 2.** Proposed resonator.

To design the resonator, the capacitance is given by Equations (1)–(6), where $t$ is the thickness (0.018 μm) of the transmission line at the substrate and $\varepsilon_0$ and $\varepsilon_r$ are, respectively, the free-space and relative permittivity ($\varepsilon_0 = 8.854 \times 10^{-12}$ F/m) [22]. $N$ is the number of turns ($N = 3$) in the split-ring structure, while $C_m$, the parasitic capacitance of the dielectric in the substrate, equaling 0.423 μF [22]. Finally, $A$ is the metal dimension of the substrate, while $d$ is the distance between the metal and the ground in the substrate [22].

$$C_{ex} = [(l_{TH} + l_{TV}) + 2\,(t + g_{ex})]C_{op}\left[\frac{\varepsilon_0(1 + \varepsilon_r)}{2}\right] \tag{1}$$

$$C_{in,m,n} = \sum_{n=1}^{N-1}\frac{C_{ex}}{4} = \frac{N-1}{2}[2l_{in,\,m,n} - (2N-1)\,(t + g_{in})]C_m\left[\frac{\varepsilon_0(1+\varepsilon_r)}{2}\right] @ C_m = \frac{\varepsilon_r A}{d} \tag{2}$$

$$C_{p1} = (l_{TH} + l_{TV})\,g_{ex} + g_{in}\left[\frac{\varepsilon_0(1 + \varepsilon_r)}{2}\right] \tag{3}$$

$$C_{p2} = (l_{in,m,n}t)\,g_{ex} + g_{in}\left[\frac{\varepsilon_0(1 + \varepsilon_r)}{2}\right] \tag{4}$$

$$C_s = 3.937 \times 10^{-5}l_T(\varepsilon_r + 1)[0.11(N - 3) + 0.252] \tag{5}$$

$$C_{op} = 3.937 \times 10^{-5}l_s(\varepsilon_r + 1) \tag{6}$$

The inductance used in designing the resonator is given by Equations (7)–(10), where $\lambda_g$, the guided wavelength, was 18.8 mm [22].

$$L_{ex} = [2(l_{TH} + l_{TV})] - g_{ex} - 2w_{ex} \tag{7}$$

$$L_{in,m,n} = 2l_{in,n} - g_{in} - 2w_{in,m,n} \tag{8}$$

$$L_T = 2 \times 10^{-4}l_T\left[log_{10}\left(\frac{l_T}{w_T + t}\right)\right] + 1.193 + 0.2235\left[\left(\frac{w_T + t}{l_T}\right)\lambda_g\right] \tag{9}$$

$$L_s = 2 \times 10^{-4} l_s \left[ log_{10} \left( \frac{l_s}{w_s + t} \right) \right] + 1.193 + 0.2235 \left[ \left( \frac{w_s + t}{l_s} \right) \lambda_g \right] \tag{10}$$

In the resonator design, the characteristic impedance was given by Equations (11)–(17), where $\omega_r$ is the resonant frequency, $\beta$ is the phase constant, and $Z_0$ is the characteristic impedance ($Z_0 = 50\ \Omega$) [22].

$$Z_{ex} = \frac{1 - \omega_r^2 2C_{ex}}{j\omega_r C_{ex}} \tag{11}$$

$$Z_{in,m,n} = \frac{1 - \omega_r^2 2C_{in,m,n}}{j\omega_r C_{in}, m, n} \tag{12}$$

$$Z_{p1} = \frac{Z_{ex}}{jsin\beta_1 + l_{ex}} \tag{13}$$

$$Z_{p2} = \frac{Z_{in,m,n}}{jsin\beta_2 + l_{in,m,n}} \tag{14}$$

$$Z_{op} = \frac{j\omega_r L_s}{1 - \omega_r^2 L_s (2C_{op})} \tag{15}$$

$$\beta_1 = \omega_r \sqrt{C_{p1}(L_{ex} + L_{in,m,n})} \tag{16}$$

$$\beta_2 = \omega_r \sqrt{C_{p2}(L_{ex} + L_{in})} \tag{17}$$

Split−ring resonators generally have high-$Q_L$ values [23]. In addition, the open-loaded T-type stub ($Z_T$ and $Z_S$) operates similar to a band-stop-type resonator with a high $Q_L$, as shown in Figure 3. From the figure, $\theta_T$ and $\theta_S$ are 30° and 45°, respectively, at the resonant frequency [24]. Then, $Z_T$ and $Z_S$ can be solved using Equations (18) and (19) for $\theta_T$.

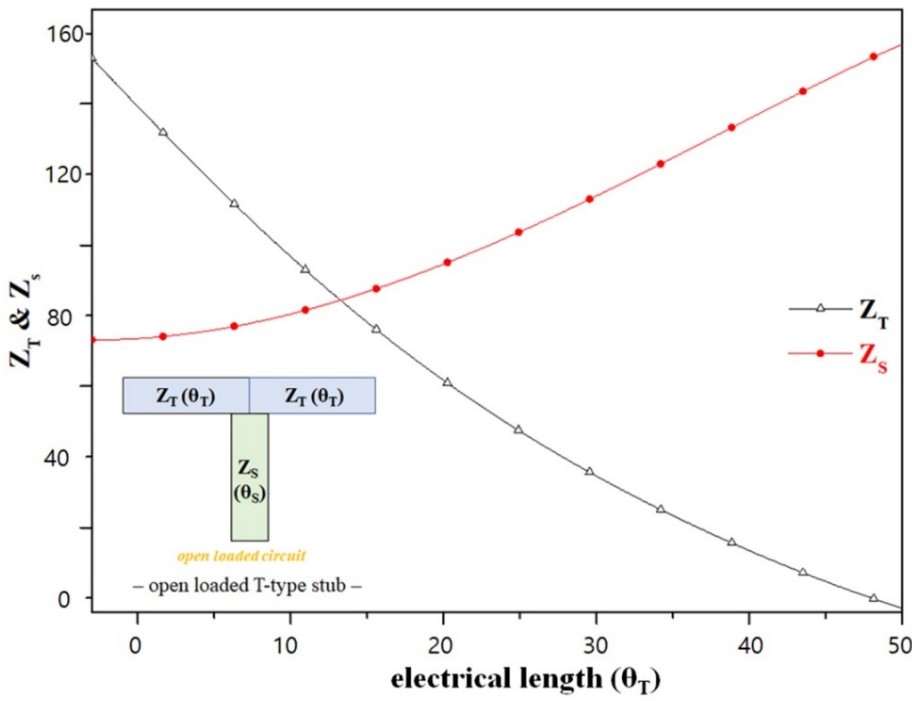

**Figure 3.** Change in electrical length according to the impedance of the stub.

$$Z_T = Z_0 cot\theta_T \tag{18}$$

$$Z_S = Z_0 \frac{cos^2\theta_T}{1 - 2sin^2\theta_T} \tag{19}$$

where $Z_0$ is characteristic impedance (50 Ω). Then, $\theta_T$ must be greater than 0° and less than 45° ($2\theta_T < \theta_T = 90°$). If $\theta_T$ is greater than 45° ($\theta_T < 90°$), $Z_S$ reaches infinity, as shown in Figure 3. For the shunt stub, $-jZ_S \cot \theta_S = \infty$; thus, the open-loaded T-type stub behaves similar to an open stub. $Z_T$ and $Z_S$ are 86.6 Ω and 75 Ω, and $\theta_T$ and $\theta_S$ are 30° and 45°, respectively.

The calculated values of the characteristic impedance, electrical, length, and physical length of the proposed resonator are listed in Table 2.

**Table 2.** Parameters of the proposed resonator.

| Parameter | | Value | Unit | Parameter | | Value | Unit |
|---|---|---|---|---|---|---|---|
| Capacitance | $C_{ex}$ | 1.86 | pF | Characteristic impedance | $Z_{ex}$ | 127 | Ω |
| | $C_{in,m,n}$ | 9.72 | pF | | $Z_{in,m,n}$ | 200 | Ω |
| | $C_{p1}$ | 8.81 | pF | | $Z_{p1}$ | 41.7 | Ω |
| | $C_{p2}$ | 9.36 | pF | | $Z_{p2}$ | 81.2 | Ω |
| | $C_S$ | 6.43 | μF | | $Z_{op}$ | 4.17 | Ω |
| | $C_{op}$ | 1.20 | mF | | $Z_T$ | 86.6 | Ω |
| Inductance | $L_{ex}$ | 2.208 | H | | $Z_S$ | 75.0 | Ω |
| | $L_{in,m,n}$ | 4.43 | H | Electrical length | $\theta_{TH}$ | 49.59° | deg |
| | $L_T$ | 11.0 | mH | | $\theta_{TV}$ | 27.27° | deg |
| | $L_S$ | 5.02 | H | | $\theta_{ex1}$ | 16.6° | deg |
| Physical length | $l_{TH}$ | 3.00 | mm | | $\theta_{ex2}$ | 8.59° | deg |
| | $l_{TV}$ | 1.65 | mm | | $\theta_{in1}$ | 32.8° | deg |
| | $l_{ex1}$ | 1.00 | mm | | $\theta_{in2}$ | 11.2° | deg |
| | $l_{ex2}$ | 0.52 | mm | | $\theta_{in3}$ | 6.72° | deg |
| | $l_{in1}$ | 2.05 | mm | | $\theta_{in4}$ | 2.08° | deg |
| | $l_{in2}$ | 0.70 | mm | | $\theta_{in5}$ | 4.64° | deg |
| | $l_{in3}$ | 0.42 | mm | | $\theta_T$ | 30.0° | deg |
| | $l_{in4}$ | 0.13 | mm | | $\theta_S$ | 45.0° | deg |
| | $l_{in5}$ | 0.29 | mm | Physical width | $w_{ex}$ | 0.19 | mm |
| | $l_t$ | 0.84 | mm | | $w_{in,m,n}$ | 0.19 | mm |
| | $l_s$ | 0.86 | mm | | $>w_T$ | 0.14 | mm |
| Gap size | $g_{ex}$ | 0.13 | mm | | $w_S$ | 0.13 | mm |
| | $g_{in}$ | 0.13 | mm | Phase constant | $\beta_1$ | 2.42 | rad/m |
| | | | | | $\beta_2$ | 2.493 | rad/m |

In the open-loaded T-type stub of the proposed resonator, the $Q_L$ started to change according to the tuning of the stub length. When the proposed resonator was designed by integrating the open-loaded T-type stub into the split-ring resonator, the $Q_L$ was 333, as shown in Figure 5, and the length of the stub was 0.82 mm. However, if the stub length is tuned, $Q_L$ varies. For example, the $Q_L$ increases (from 83.0 to 583) in the stub length range of 0.78 mm to 0.86 mm, and it gradually decreases (from 500 to 98.7) thereafter.

When the open-loaded T-type stub is directly connected to the split-ring resonator, the $Q_L$ of the resonator is greatly increased. Figure 4 shows the results of the simulation of the $Q_L$ of the split-ring resonator (single), open-loaded T-type stub, and integrated split-ring resonator (single) with an open-loaded T-type stub. From the figure, the $Q_L$ values of the split-ring resonator (single), open-loaded T-type stub, and integrated split-ring resonator (single) with the open-loaded T-type stub were 206 ($S_{21}$: 24.8 dB), 322 ($S_{21}$: 37.4 dB), and 323 ($S_{21}$: 56.6 dB), respectively.

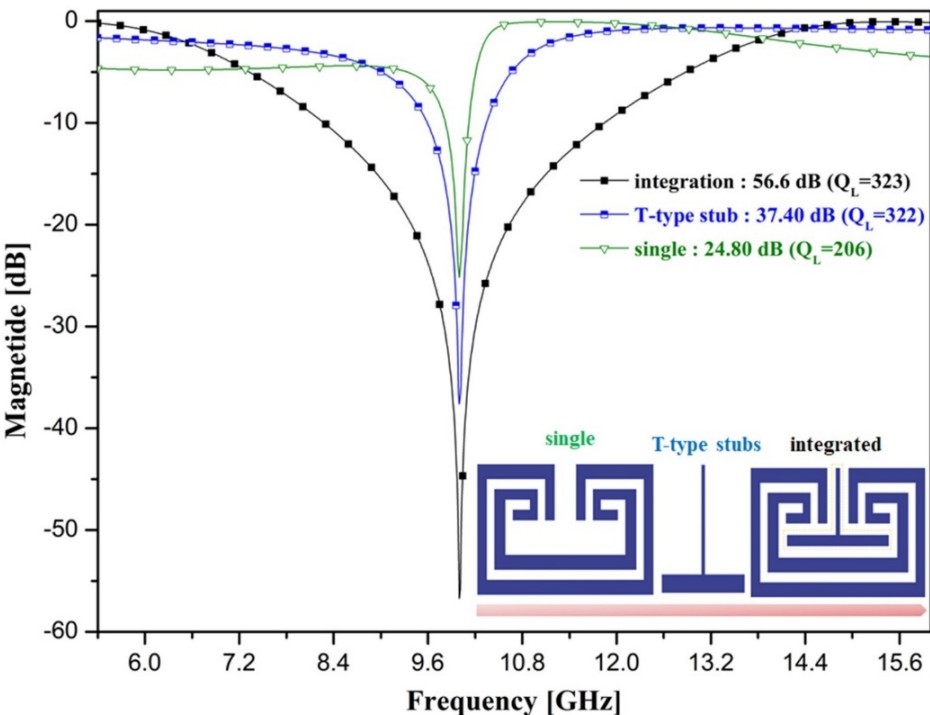

**Figure 4.** $Q_L$ change process in the resonator design stage.

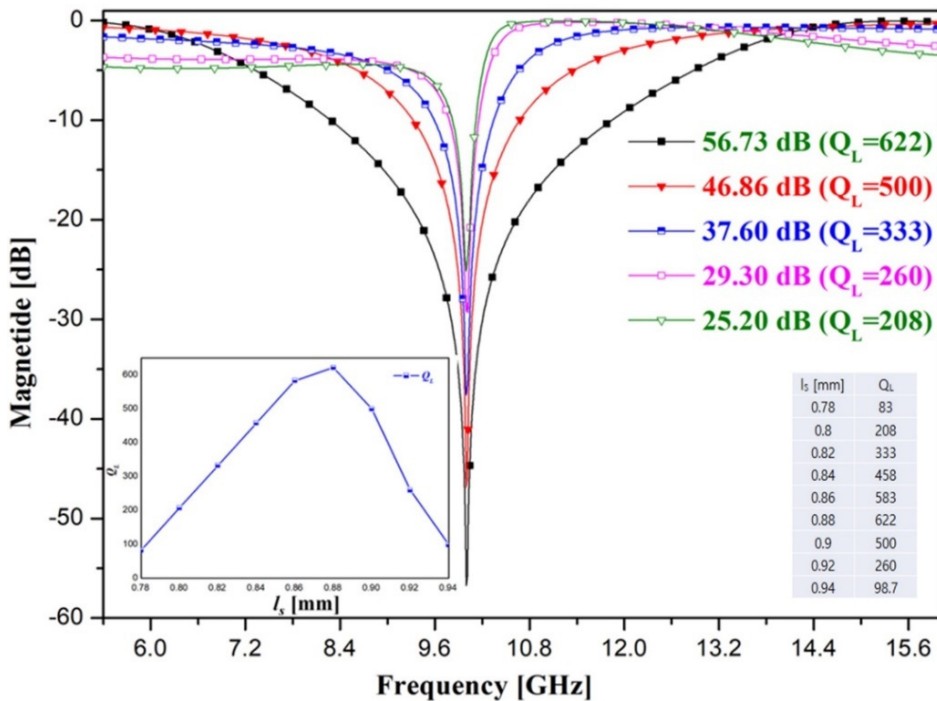

**Figure 5.** $Q_L$ change according to stub length adjustment.

The fabrication of the proposed resonator is shown in Figure 6. Specifically, the fabricated resonator employs a Teflon substrate with a low dielectric constant of 2.54, height of 0.54 mm, and thickness (*t*) of 0.018 μm, and the size of the resonator is 3.0 mm × 1.65 mm. The simulated value of $Q_L$ is 683 at 10.0 GHz for this resonator, and the measured result is 632 at 10.008 GHz.

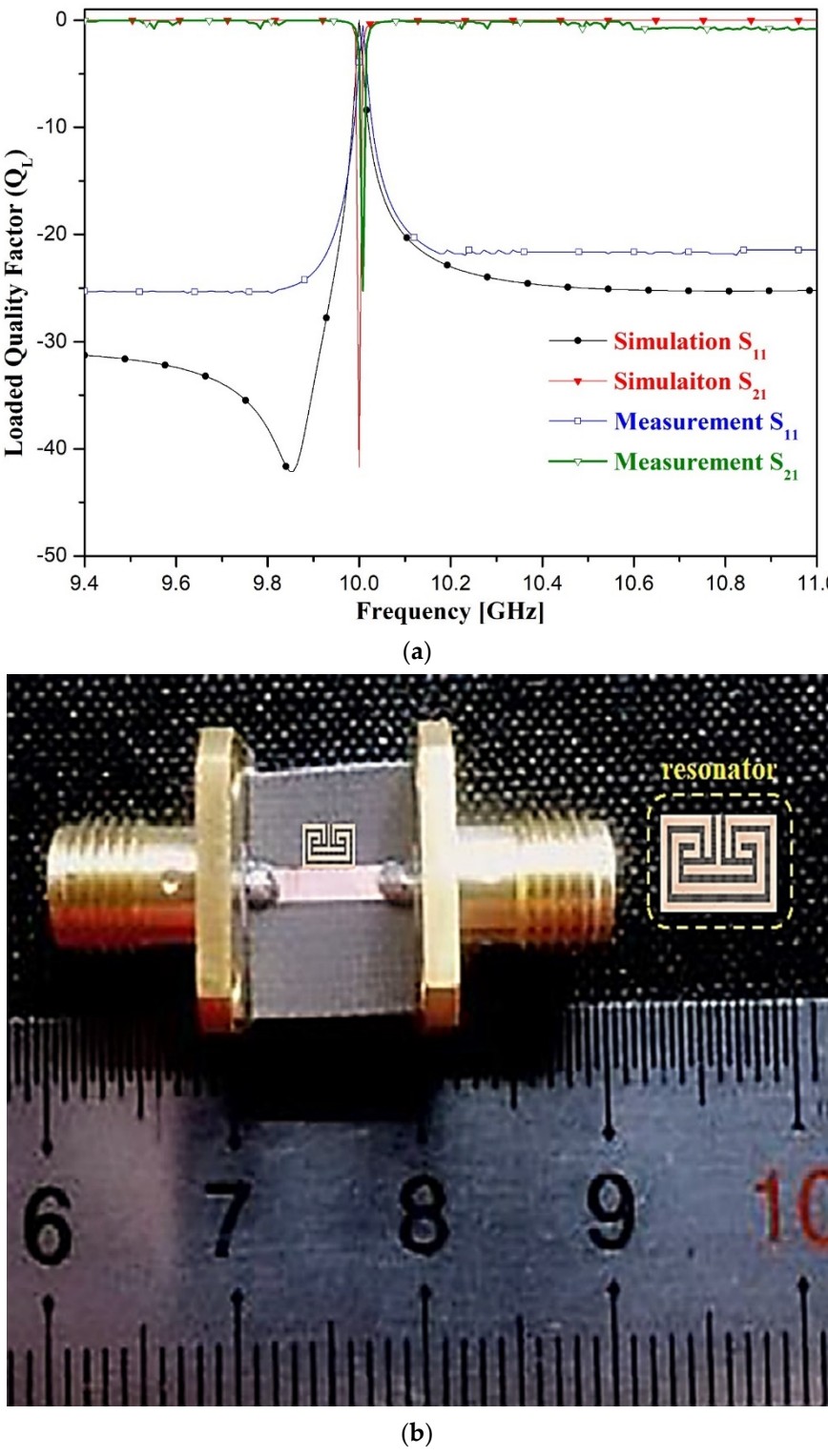

**Figure 6.** Fabrication and experimental results of the proposed resonator: (**a**) simulation and measurement results; (**b**) fabrication.

## 4. Oscillator Design and Measurement Results

In this study, a low-phase-noise X-band oscillator was designed using a new high-$Q_L$ resonator, as shown in Figure 7. As shown in the figure, an oscillator was used for the series feedback structure and self-bias method, and the series feedback at the source was chosen as an oscillator, which provides the negative resistance ($-R$) to the gate of a GaAs-MESFET.

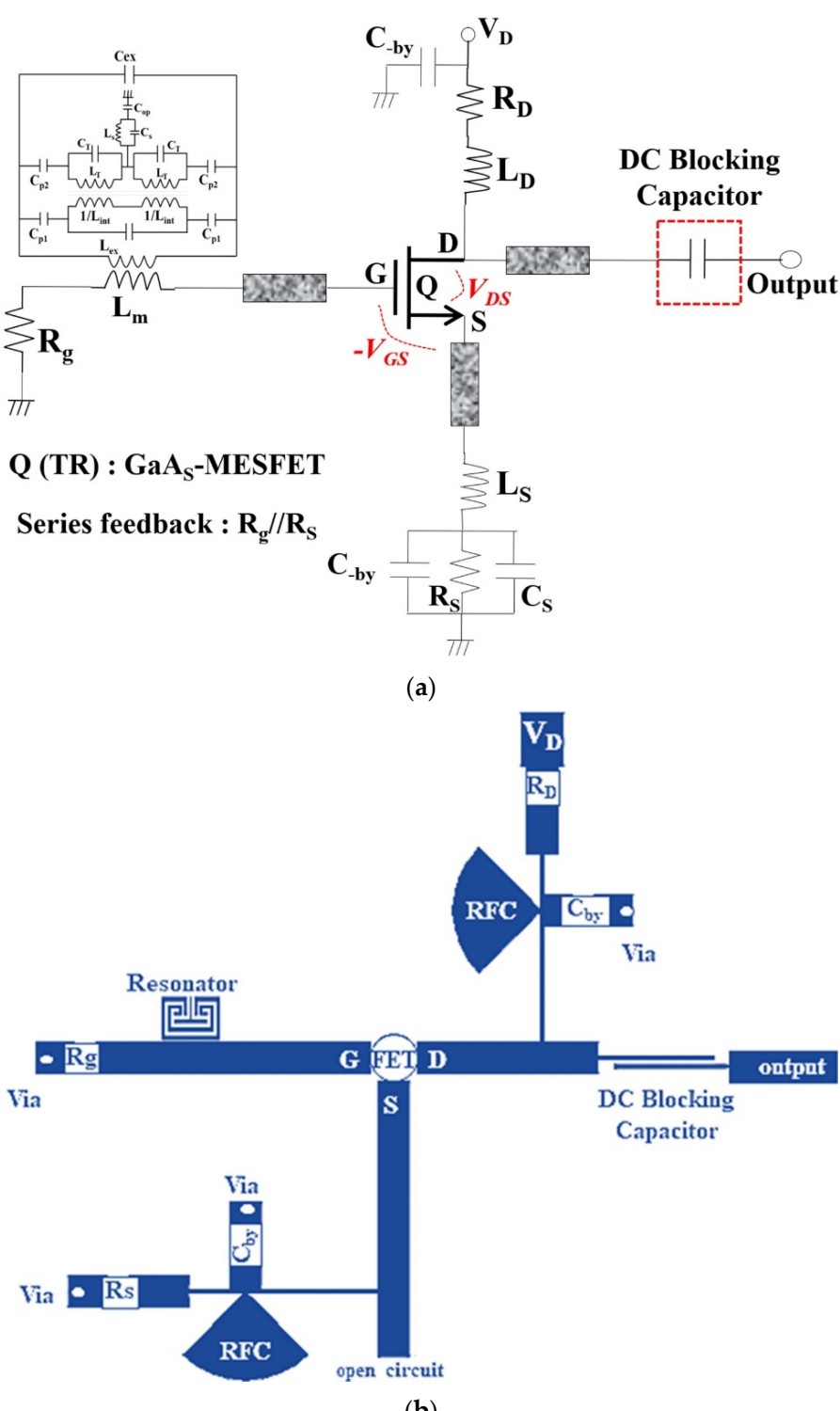

(**a**)

(**b**)

**Figure 7.** *Cont.*

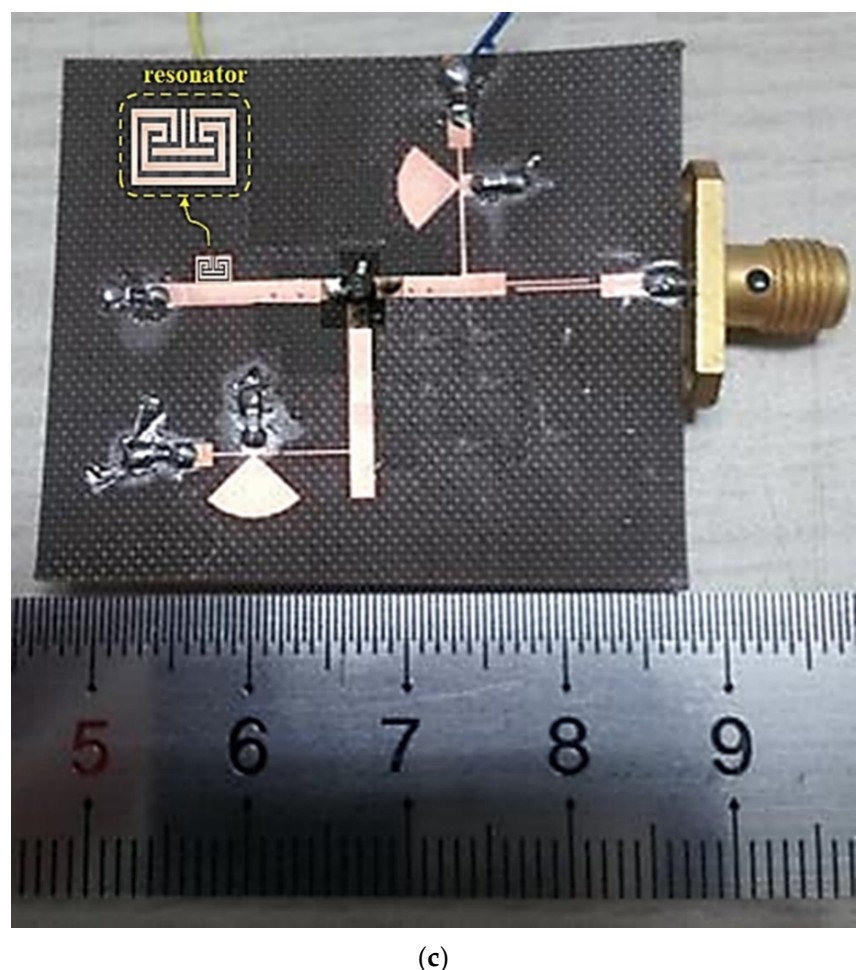

**(c)**

**Figure 7.** Designed oscillator: (**a**) equivalent circuit, (**b**) structure (layout), (**c**) fabrication.

The oscillator is composed of a $\lambda_g/4$ transmission line (microstrip line) and a radial stub for a DC-bias circuit providing RF power to a wideband open-loaded circuit. The proposed high-$Q_L$ resonator is connected to the gate (G) of the GaAs-MESFET. $V_D$, $R_D$, and $Rg$ are the drain (D) bias voltage and drain (D) resistance; further, $R_S$ and $C_S$ are the source (S) resistance and source capacitance, respectively.

$L_D$ and $L_S$ are the RF-chock (RFC) in the drain (D) and source (S), respectively. Radial stubs are the RFC circuits in the drain (D) and source (S) loads. The gate (G) bias is self-biased, and a capacitor ($C_{\_by}$) is used only for by-pass C for the bias. The fabricated oscillator was used as the Teflon substrate, and the dielectric constant ($\varepsilon_r$), height (h), and thickness (*t*) of the Teflon substrate were 2.55, 0.54 mm, and 0.018 mm. The fabrication was performed by wet etching, which is used for negative film development.

The measurement results of the new oscillator are shown in Figure 8, in which the oscillation frequency is 10 GHz. The phase noise of the oscillator is observed to be −138.13 dBc/Hz at an offset frequency of 100 kHz. Thus, the $Q_L$ of the resonator has a considerable effect on the phase noise performance of the oscillator. Additionally, the output power and amount of second harmonic suppression for the X-band oscillator are 18.66 dBm and −34.40 dBc under bias conditions of $V_{DS}$ = 3 V and $I_D$ = 40 mA, respectively. The figure of merit (FOM) is given by Equation (20), which is −195.7 dBc/Hz at a 100 kHz offset, where the $L\{f_{offset}\}$ is the phase noise in the dBc/Hz at the offset frequency, and $f_{offset}$ is from the carrier (oscillation) frequency, $f_0$. $P_{DC}$ is the DC power dissipation in mW [13].

$$\text{FOM} = -20log\left(\frac{f_0}{f_{offset}}\right) + L\left\{f_{offset}\right\} + 10log\left(\frac{P_{DC}}{1\text{ mW}}\right) \qquad (20)$$

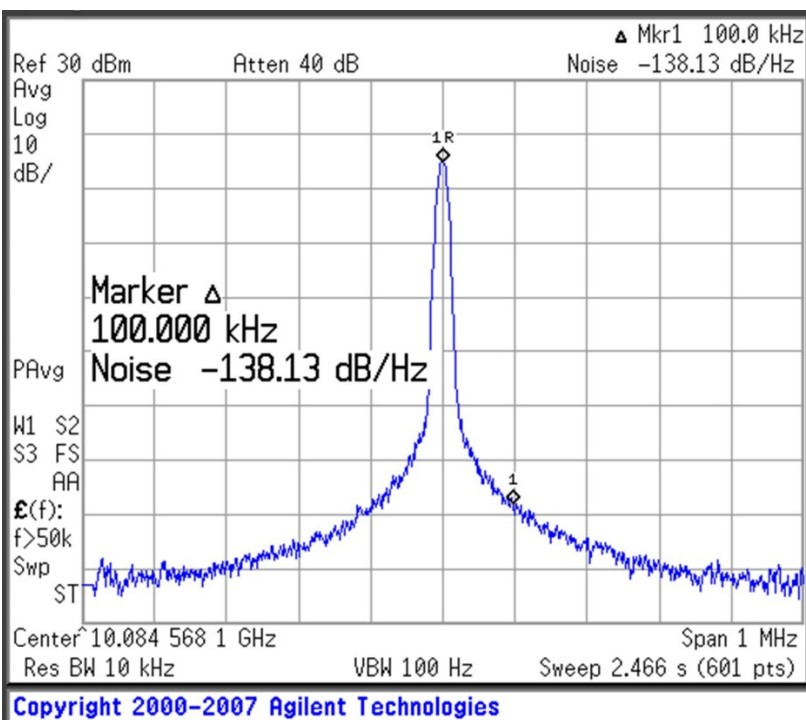

(**a**)

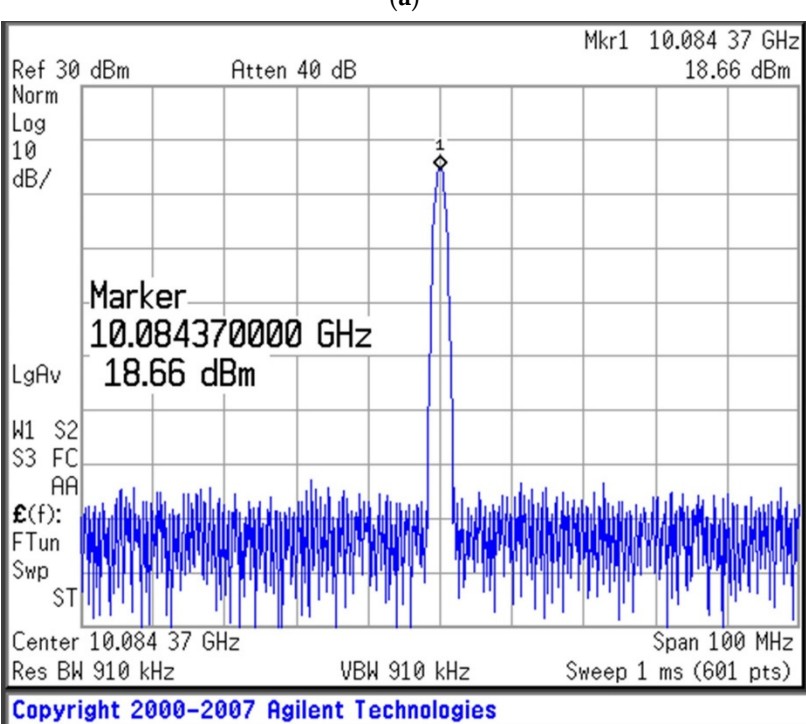

(**b**)

**Figure 8.** *Cont.*

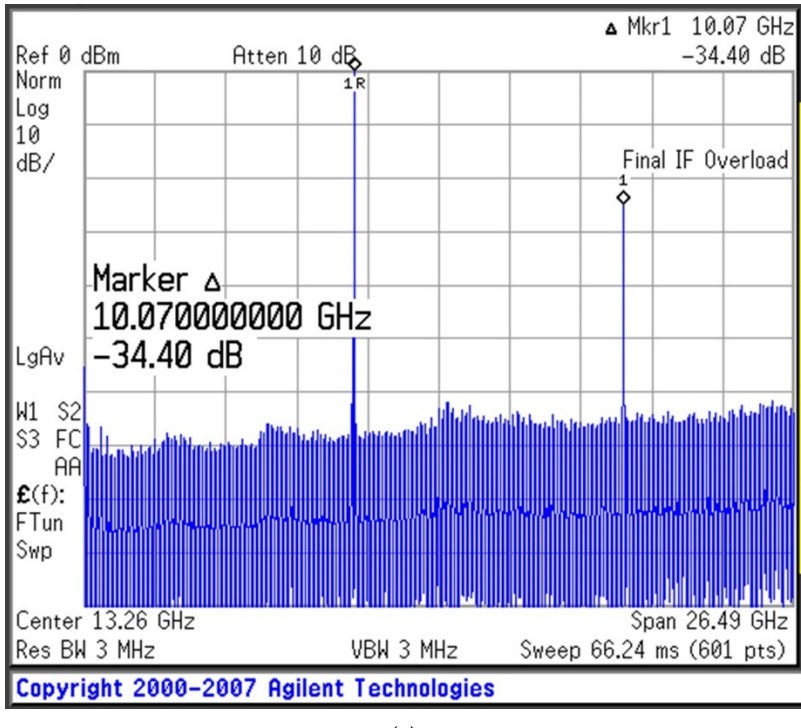

(**c**)

**Figure 8.** Experimental results for the designed oscillator ($V_{DS}$ = 3 V, $I_D$ = 40 mA): (**a**) phase noise, (**b**) output power, and (**c**) second harmonic suppression.

An Agilent E4440A PSA series spectrum analyzer was used to measure the oscillations. The phase noise and output power of the fabricated oscillator were compared with those of oscillators developed in previous studies, as shown in Table 3.

**Table 3.** Comparison between the parameters of the proposed oscillator and those of oscillators developed in previous studies.

| Reference | Frequency (GHz) | Phase Noise (dBc/Hz) at 100 kHz | Output Power (dBm) | $Q_L$ | FOM (dBc/Hz) at 100 kHz |
|---|---|---|---|---|---|
| This work | 10.08 | −138.13 | 18.66 | 632 | −195.7 |
| [10] | 10.98 | −121.6 | 1.800 | − | −182.3 |
| [11] | 10.00 | −95.40 | 10.16 | 190 | −135.3 |
| [12] | 9.883 | −97.60 | 3.020 | 260 | −182.2 |
| [13] | 9.010 | −104.3 | 1.800 | 104 | −104.3 |
| [14] | 10.11 | −108.7 | 4.600 | 520 | −172.2 |
| [15] | 9.850 | −124.8 | 3.600 | 243 | −189.1 |
| [16] | 6.200 | −104.62 | 14.68 | 430 | −156.2 |
| [17] | 9.960 | −128.30 (@ 1 MHz) | 8.570 | 66.7 | −193.2 (@ 1 MHz) |
| [20] | 8.080 | −109.94 | 2.140 | − | −174.2 |
| [21] | 8.172 | −112.0 | 4.000 | − | −174.2 |

Offset frequency: 100 MHz [17].

## 5. Conclusions

In order to find the location of the marker, the existing method uses radiography-based fluoroscopic radiography to find the location through images [25]. However, the fluoroscopic radiography is large and heavy. In addition, fluoroscopic radiography has a high unit price. Therefore, the heavy size will induce difficulty in performing surgery. However, sensors to find markers are low in unit price and light in weight; the sensor is also small in size. Therefore, it is easy to handle during surgery. The most important thing in these

sensors is the oscillator. Since the oscillator has a very low phase noise, it is considered to have excellent performance.

The designed oscillator consists of a plane shape. The resonator has an open structure, and this resonator has not applied a via design method. Therefore, the resonator has a high loaded quality factor ($Q_L$) because there is no loss of concentration of via energy [26,27]. When applying the proposed oscillator to a sensor, if the amplifier is connected, it can be used as a radar-based sensor with high performance, which would be very useful in finding markers installed on tumors. The reason is that markers are made of metal. When a signal generated from the oscillator is reflected through a metal marker, the sensor may detect a reflected signal to locate the tumor. Therefore, it will be important that the performance of the sensor to locate the tumor suppress the phase noise of the oscillator.

In this paper, a low-phase-noise-X-band oscillator with a high-$Q_L$ resonator using an open-loaded T-type stub and split-ring structure is presented. The proposed oscillator has a high $Q_L$ and low phase noise because it is a resonator in which an open-loaded T-type stub and a split-ring structure are combined. The proposed oscillator can increase the $Q_L$ of a resonator by adjusting the length of an open-loaded T-type stub; thus, the phase noise of the oscillator can be sufficiently reduced. In addition, the resonator is very small. The oscillator has a high output power and can perform second harmonic suppression. The measurement results indicate that an output power of 18.66 dBm and second harmonic suppression of $-34.40$ dBc (at 13.26 GHz) can be realized. At an operating frequency of 10.084 GHz, the phase noise is $-138.13$ dBc/Hz at a 100 kHz offset. This low-phase-noise X-band oscillator can be fabricated with a monolithic microwave integrated circuit (MMIC) technique, owing to its entirely planar structure. It can be applied to sensors to detect the location of tissues during laparoscopic surgery.

**Author Contributions:** Design and simulation; K.-C.Y., analysis and supervisor; K.-G.K. and J.-W.C., measurement; B.-S.K. All authors have read and agreed to the published version of the manuscript.

**Funding:** This research was supported by the Gachon University Gil Medical Center (Grant number FRD2019-08) and by the GRRC program of the Gyeonggi province (No. GRRC-Gachon2020(B01). In addition, the research work was supported by the Institute of Information & Communications Technology Planning & Evaluation (IITP) grant funded by the Korean government (MSIT) (No. 2020-0-00161-001, Active Machine Learning based on Open-set training for Surgical Video).

**Institutional Review Board Statement:** Not applicable.

**Informed Consent Statement:** Not applicable.

**Data Availability Statement:** The data presented in this study are available upon request from the corresponding author. The data are not publicly available because of privacy and ethical restrictions.

**Acknowledgments:** This device was used for the fabrication and measurements thanks to Tae-Hyeon Lee at the Gyeonggi University of Science and Technology in Siheung, Republic of Korea.

**Conflicts of Interest:** The authors declare no conflict of interest.

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
