# Peer review of "Low-Phase-Noise Oscillator Using a High-QL Resonator with Split-Ring Structure and Open-Loaded T-Type Stub for a Tumor-Location-Tracking Sensor"

_applsci, doi:10.3390/app112311550_

Round 1

Reviewer 1 Report

Line 47: "should be of a low phase" can be rephrased.

Line 93: -100 dBc/Hz 

Line 131: Is 0.423 F correct?

Authors should describe in details what two ports of the equivalent circuit is representing physically in the layout.

Line 133 to 154: Please, add references for the equations. Or describe derivation of major equations.

Line 156: QL -> Q with subscript of L

Line 218: Missing gate (G) resistance

The power of the oscillator is quite high. I think it would be better to add FOM in Table 3.

Authors are mentioning second harmonic suppression in conclusions. Please, add wideband frequency response figure in the body to show the response at the second harmonic frequency.

Author Response

Point 1: Line 47: "should be of a low phase" can be rephrased.

Response 1: Thank you very much for your kindly comments. I changed the “can be” instead of should be. Please refer to line 47 with red words.

Point 2: Line 93: -100 dBc/Hz

Response 2: Thank you for your good comments. I modified the -100dBc/Hz. Please refer to line 94 with red words.

Point 3: Line 131: Is 0.423 F correct?

Response 3: It's not "F". μF is right. I made a correction. Refer to the 135st line, red sentence.

Point 4: Authors should describe in details what two ports of the equivalent circuit is representing physically in the layout.

Response 4: Figure 2 was modified (equivalent circuit and resonator structure). And I explained the meaning of port by adding the sentence of 127-130. Please check the red sentence.

Point 5: Line 133 to 154: Please, add references for the equations. Or describe derivation of major equations.

Response 5: Reference [22] has been added to the 131-159 line.

Point 6: Line 156: QL -> Q with subscript of L

Response 6: Thank you for the typo. I modified the content (QL). Please check the red number 160.

Point 7: Line 218: Missing gate (G) resistance

Response 7: Fixed all the same with 'Rg' instead of 'RG'. The figure has also been modified with Rg. The reason I modified it to Rg is because it is commonly used in textbooks. thank you. Please refer to 222 line and red color.

Point 8: The power of the oscillator is quite high. I think it would be better to add FOM in Table 3.

Response 8: I added the FOM value and equation (2). Please refer to line 235-240 of red words and Table 3.

Point 9: Authors are mentioning second harmonic suppression in conclusions. Please, add wideband frequency response figure in the body to show the response at the second harmonic frequency.

Response 9: The frequency band of 13.26 GHz is marked for the second harmonic. Please check line 299 and the red sentence.

Reviewer 2 Report

The proposed oscillator shows good noise results compared to other previous devices. However, there are some points that I want to comment

  1. Consider editing the abstract and the introduction part in order to make the paper easier to understand and appeal your results.
  2. Discussion part can be improved.
  3. Please comment about the advantage and the effectiveness of the proposed oscillator for the tumor location tracking when compared to other non-invasive imaging systems such CMOS imager.
  4. Proofreading is advised to fix some errors.

Author Response

The proposed oscillator shows good noise results compared to other previous devices. However, there are some points that I want to comment.

Point 1: Consider editing the abstract and the introduction part in order to make the paper easier to understand and appeal your results.

Response 1: I checked the summary and introduction once again. There was a part of the sentence that was not understood. Therefore, I revised the sentence once again to make it understandable.

These are the blue ones in the introduction. Please check lines 39, 41, 43-44, 46-49, 57, 73-74.

Thank you.

Point 2: Discussion part can be improved.

Response 2: In the conclusion, I added the consideration. Please check the blue sentence on line 282-290. Thank you.

Point 3: Please comment about the advantage and the effectiveness of the proposed oscillator for the tumor location tracking when compared to other non-invasive imaging systems such CMOS imager.

Response 3: We compared and considered the proposed oscillator with the existing system. And I supplemented the advantages. Please refer to line 274-281 in green sentence.

Point 4: Proofreading is advised to fix some errors.

Response 4: I checked the entire sentence error. And I corrected English through an expert. Thank you for pointing out.
